# The Journey of Sentinel Lymph Node Biopsy in Cutaneous Melanoma: A Brief Narrative Review from Scalpel to Smart Tech

**DOI:** 10.3390/medicina61091542

**Published:** 2025-08-27

**Authors:** Rǎzvan Ioan Andrei, Silviu Cristian Voinea, Cristian Ioan Bordea, Aniela Roxana Nodiți, Teodora Mihaela Peleașă, Alexandru Blidaru

**Affiliations:** 1Department of General Surgery, “Carol Davila” University of Medicine and Pharmacy, B-dul Eroii Sanitari 8, 050474 Bucharest, Romania; ioan-razvan.andrei@umfcd.ro (R.I.A.);; 2Department of Surgical Oncology, Institute of Oncology “Prof. Dr. Al. Trestioreanu”, Şos. Fundeni 252, 022328 Bucharest, Romania; 3Department of Plastic and Reconstructive Surgery, Prof. Dr. “Agrippa Ionescu” Hospital, Str. Arhitect Ion Mincu 7, 011356 Bucharest, Romania

**Keywords:** cutaneous melanoma, sentinel lymph node biopsy, augmented reality

## Abstract

Sentinel lymph node biopsy (SLNB) has transformed the management of cutaneous melanoma, emerging as a cornerstone in evaluating regional lymphatic spread while minimizing surgical morbidity. From its theoretical foundation laid by Cabanas to its refinement and clinical validation through landmark trials, SLNB has evolved into a standard of care with significant prognostic value. This review traces the historical trajectory of SLNB, analyzes current guidelines and controversies and explores future directions. Novel imaging technologies, such as indocyanine green fluorescence and augmented reality-assisted mapping, promise to enhance accuracy and reduce invasiveness. Furthermore, the advent of effective systemic therapies and neoadjuvant protocols is reshaping the therapeutic landscape, potentially redefining the role of SLNB in melanoma management. As precision medicine advances, SLNB remains an essential procedure, with its utility continually redefined by technological innovation and evolving oncologic strategies.

## 1. Introduction

The development of sentinel lymph node biopsy (SLNB) marked a significant shift by allowing surgeons to detect microscopic metastatic disease with minimal invasiveness, greatly reducing patient morbidity compared to traditional elective lymph node dissections (ELND) [1].

Since its initial conceptualization and subsequent refinement, SLNB has evolved into the standard procedure for assessing regional nodal status in melanoma patients [2]. Landmark studies have solidified the role of SLNB by demonstrating its prognostic significance and shaping current clinical practice guidelines [3].

Beyond its prognostic significance, SLNB is continually evolving in response to technological advances and emerging clinical needs. Innovations such as fluorescence-guided mapping with indocyanine green (ICG), dual-tracer techniques and the exploration of augmented reality (AR) for surgical navigation are refining accuracy and reducing invasiveness [4,5,6]. In parallel, the advent of systemic therapies, including immune checkpoint inhibitors and targeted therapies, has initiated a paradigm shift that raises important questions about SLNB’s therapeutic role in the era of effective adjuvant and neoadjuvant interventions [7,8].

This paper aims to provide a short overview of the historical development of SLNB, the current evidence-based standards and exciting future perspectives. As we enter an age of precision medicine, where artificial intelligence (AI) continues to expand the diagnostic landscape, SLNB remains a cornerstone of melanoma management—yet its application, value, and methodology are being actively redefined. Understanding this dynamic evolution is essential for optimizing outcomes and ensuring that patient care remains at the forefront of oncologic innovation.

## 2. Materials and Methods

To compile a comprehensive and representative overview of the evolution of SLNB from its inception to emerging smart technologies, we performed a structured literature search in major biomedical databases, namely PubMed, Scopus, Web of Science, and Medline. The search strategy combined the keywords “sentinel lymph node biopsy”, “SLNB”, “melanoma”, “cutaneous melanoma”, “indocyanine green”, “ICG”, “blue dye”, “radioactive tracer”, “Tc99m”, “augmented reality”, “artificial intelligence”, and “liquid biopsy” in various combinations, using Boolean operators to ensure inclusion of the widest possible range of relevant studies.

Inclusion criteria comprised articles in English published between April 1992 and June 2025, reflecting both the historical milestones of SLNB and advances in diagnostic and intraoperative technologies. Articles were considered eligible if the search terms appeared in the title or abstract and if they focused primarily on SLNB in the context of cutaneous melanoma, with secondary inclusion of studies describing technical adaptations in other malignancies.

We excluded conference abstracts without full-text availability, publications in languages other than English, and studies unrelated to surgical or technological aspects of SLNB. Older landmark studies predating 1992 were included selectively if they represented pivotal steps in the conceptual development of SLNB (e.g., Snow 1892, Cabanas 1977, [9,10]).

The synthesis followed a narrative review format, structured chronologically and thematically to reflect the “scalpel to smart tech” journey. References were managed using EndNote X9, and duplicate entries were removed prior to final analysis.

## 3. Content

### 3.1. A Historical Overview of Sentinel Lymph Node Biopsy

The importance of regional lymph node status and their surgical treatment in melanoma dates back to more than a century. Herbert Snow, a surgeon practicing in London in the late 19th century, meticulously documented several melanoma cases [9]. He highlighted melanoma’s tendency to spread silently to regional lymph nodes and proposed proactive surgical removal of these lymph nodes to intercept the disease early, thereby preventing further metastatic spread [9,11]. Snow argued that delaying lymph node removal until they became clinically noticeable allowed the disease to advance, significantly worsening patient prognosis [9]. This early surgical approach eventually evolved into what we now recognize as ELND. While the clinical benefits of ELND remained controversial, studies consistently showed that patients with microscopically detected nodal invasion experienced improved survival compared to patients whose lymph node involvement was diagnosed only when it became clinically palpable [12,13]. However, since most melanoma patients present without nodal metastasis at diagnosis, they gain no therapeutic advantage from such preventive lymph node removal [12]. Additionally, ELND is associated with increased morbidity, including lymphedema [14].

The major trials addressing this issue, before the 2000s, revealed no substantial survival differences among immediate and delayed lymphadenectomy, just a slight survival advantage in the ELND group for early-stage with good prognostic factors like non-ulcerated, thinner tumors and extremity melanomas [15,16,17,18]. The introduction of SLNB dramatically reshaped melanoma management, with each successive breakthrough consolidating its role (Figure 1). This technique was initially introduced by Ramon M. Cabanas in 1977. Through his studies on penile cancer, Cabanas observed that lymphatic drainage follows a consistent and predictable pathway, enabling identification of the first lymph node that directly receives lymphatic drainage [10]. He suggested in his publication that detecting metastases in this sentinel node could determine whether a more extended lymph node dissection was necessary [10].

The concept was extended and refined in 1992 by Donald L. Morton and his team from the University of California [19]. They demonstrated that sentinel lymph node biopsy could be effectively applied in cutaneous melanoma to detect metastases at an early stage and marked a crucial shift from anatomical to physiological lymphatic assessment [19]. While earlier studies had identified fixed anatomical sentinel nodes in other cancers, Morton’s team innovatively determined the sentinel node using lymphatic mapping—a functional, individualized approach [19]. Their work originated from early lymphoscintigraphy was used initially to define lymph node basins needing ELND, particularly in patients with ambiguous drainage patterns [19]. Over time, advances in tracer agents and imaging revealed that melanoma cells typically migrate to a limited number of nodes, prompting Morton and his colleagues to hypothesize that examining these select nodes could accurately reflect the status of the entire nodal basin and finally led to the adoption of SLNB as a minimally invasive procedure [19,20].

The Multicenter Selective Lymphadenectomy Trial (MSLT-I) was a major international prospective randomized study involving multiple centers. Participants were randomized into two groups: one receiving SLNB and completion lymph node dissection (CLND) if metastases were found in the sentinel node and the other with nodal observation and delayed CLND. MSLT-I demonstrated that sentinel lymph node status significantly predicted patient outcomes [21]. Importantly, the trial showed a clear benefit in recurrence-free survival for patients undergoing SLNB, both in intermediate and thicker melanoma groups [22]. Regardless of any direct survival benefit, MSLT-I firmly established the prognostic value of SLNB, facilitating its widespread acceptance and integration into melanoma management guidelines.

Initially, SLNB guided decisions primarily around performing CLND. However, accumulating experience revealed that most of these patients had no additional nodal disease, raising doubts about the necessity of routine CLND due to its considerable morbidity—particularly in groin dissections, where complications reached 51%, and lymphedema rates were around 32% compared to significantly lower rates with SLNB alone [14]. The Multicenter Selective Lymphadenectomy Trial (MSLT-II) and the German Dermatologic Cooperative Oncology Group’s Sentinel Lymph Node Trial (DeCOG-SLT), directly evaluated whether CLND could safely be omitted in patients with positive sentinel nodes. Neither study found substantial benefits regarding their primary outcomes—melanoma-specific survival (MSLT-II) or distant metastasis-free survival (DeCOG-SLT)—though MSLT-II noted improved disease-free survival mainly driven by reduced nodal recurrence [23,24,25,26]. Ultimately, these influential trials prompted a shift toward nodal observation using ultrasound surveillance as standard practice for sentinel node-positive melanoma patients. This has led to a 70% decrease in the number of CLNDs, with the current indication being limited to cases with multiple positive sentinel lymph nodes, high tumor burden, or inability to ensure adequate follow-up [23] (Table 1).

SLNB has evolved significantly since its introduction, establishing itself as a standard procedure. Traditionally, SLNB was recommended for patients with intermediate-thickness melanomas, specifically those with a Breslow index between 1.0 mm and 4.0 mm. This recommendation was based on studies indicating an unacceptable higher likelihood of regional lymph node metastasis at a Breslow index higher than 4.0 mm [27,28]. For tumour thickness less than 1.0 mm, SLNB was not routinely recommended because the possibility of sentinel node metastasis is low and the risks of the procedure outweigh the potential benefits [28].

### 3.2. Current Standards

According to current clinical guidelines, SLNB is recommended in patients with cutaneous melanoma and a Breslow index greater than 0.8 mm, in the context of clinically and radiologically negative regional lymph nodes (N0) and no evidence of distant metastasis (M0) [29]. This indication remains valid even in cases with a Breslow index exceeding 4.0 mm, where the risk of distant metastasis may be higher [29,30]. In these patients, SLNB continues to provide critical prognostic information, ensures accurate pathological staging, and may guide decisions regarding adjuvant systemic therapy or inclusion in clinical trials targeting high-risk populations. In the subset of thin melanomas, SLNB may be considered on an individual basis for lesions with Breslow thickness between 0.5 mm and 0.8 mm, particularly when unfavorable histopathological features are present and is associated with an increased risk of sentinel lymph node metastasis, justifying the performance of SLNB even in melanomas traditionally considered low risk [31].

Broad consensus supports the recommendation of SLNB for patients whose estimated risk of nodal metastasis exceeds 10%, whereas those with less than a 5% risk typically do not require the procedure. Patients with an intermediate risk (5–10%) are best managed through shared decision-making processes [32,33]. Online calculators developed by the Memorial Sloan Kettering Cancer Center and the Melanoma Institute Australia are being used and rely on prognostic factors such as tumor thickness, ulceration status, patient age, mitotic rate, Clark’s level of invasion, lymphovascular invasion, anatomical tumor location and histological subtype [34,35].

Patients with previous local excisions pose unique challenges due to disrupted lymphatic drainage, which can lower the accuracy of SLNB, so current recommendations are that diagnosis should be based on a full thickness complete excision with a minimal margin of clinically uninvolved skin, to permit accurate subsequent lymphatic mapping [36]. If necessary, additional re-excision of the scar would be performed to obtain oncological resection margins (1 to 2 cm for invasive and 0.5 for in situ) [37]. The timing of SLNB for cutaneous melanoma is recommended at the same time as the wide local excision (WLE) [29].

### 3.3. Standardization Challenges: Variability in SLNB Protocols

Blue dye is not universally used for sentinel lymph node biopsy due to several limitations and regional considerations. One major factor is the risk of allergic reactions, including rare cases of severe anaphylaxis, which has led some health systems to restrict or avoid its use altogether [38]. Additionally, blue dye can cause temporary skin staining and interfere with postoperative wound assessment. In some countries, regulatory approval for certain dyes, such as patent blue V or isosulfan blue, is limited or absent, making them unavailable for clinical use.

Dual-tracer techniques combining radioactive tracer with blue dyes enhance accuracy and reduce false-negative rates [39]. However, the radioactive tracer method alone is highly spread in clinical practice because it still offers a high identification rate and is free from allergic reactions. The drawbacks are radiation exposure, logistical challenges (including the necessity of having a nuclear medicine department), and potential delays in surgical workflow.

The use of ICG fluorescent dye as an alternative to traditional radiotracer methods for SLNB in cutaneous melanoma patients was evaluated and the findings suggest that ICG fluorescence imaging is a reliable and potentially more cost-effective alternative [40].

However, even if ICG-guided SLNB is comparable to blue dye in terms of detection rate and SLN positivity, it can not be used alone to identify all positive SLNBs [41].

Unfortunately standardization is hampered by difference in regulatory approval, substance availability and surgeons preferences in regarding to dose, injection site and timing.

### 3.4. Future Horizons

The recent approval of systemic adjuvant therapies for Stage IIB and IIC melanoma has put into question the continuing role of SLNB in these high-risk, node-negative patients. Despite systemic treatment availability, SLNB provides crucial prognostic information, significantly affecting melanoma-specific survival predictions and subsequently guiding adjuvant treatment decisions. For now, SLNB continues to hold significant clinical value even in the evolving landscape of systemic therapies [25,42,43].

The future of SLNB may follow two potential trajectories: the evolution and transformation of the technique or even its omission. However, these paths are not mutually exclusive. Instead, they appear to converge in specific clinical contexts.

Looking ahead, the incorporation of neoadjuvant therapies such as checkpoint inhibitors may significantly alter the surgical approach to melanoma management. Early results from trials like PRADO suggest the potential for a more personalized, less invasive surgical strategy following neoadjuvant treatment [42,43]. Although these strategies have not yet conclusively demonstrated superiority regarding recurrence-free survival endpoints, they represent promising areas for ongoing research and potential shifts toward less invasive or even non-surgical management in selected patient subsets [44,45].

Melanin pigmentation, beyond being a marker of melanocytic origin, directly influences melanoma biology by shaping tumor metabolism, oxidative balance, immune interactions and can attenuate the effects of radio-, chemo-, and immunotherapy, making melanogenesis not only a diagnostic feature but also a therapeutic challenge in melanoma management [46]. Increased pigmentation in melanomas has been correlated with shorter disease-free and overall survival, as melanin and its intermediates can generate pro-oxidative, mutagenic, and immunosuppressive environments that promote tumor growth and therapy resistance [47]. Implementing routine evaluation of melanin pigmentation and developing strategies to modulate melanogenesis hold significant promise for more personalized and effective treatment strategies.

In 2023, Peter A. von Niederhäusern from the Department of Biomedical Engineering, the University of Basel, demonstrated a proof of concept for an AR approach for SLNB for oral and oropharyngeal squamous cell carcinoma which showed promising results [48]. Even though it is still difficult to implement and has its limits, AR technology is increasingly explored for minimally invasive surgery, particularly in orthopedics and laparoscopic procedures. Current research primarily investigates AR-based intraoperative navigation, education and training. Despite experimental evidence suggesting shorter surgery times and reduced complications, AR systems have not yet consistently outperformed traditional navigation methods in accuracy. Challenges remain, such as reliable patient-data registration, computational limitations, and limited surgeon awareness. Future developments, supported by innovations in robotics and AI, could enhance AR’s clinical utility and training effectiveness [49].

Recent advances in AI have shown great promise in transforming pathology by enhancing diagnostic accuracy, reproducibility, and prognostic predictions. Despite current challenges in clinical integration—such as interpretability, validation, and standardization—the synergy between AI and human expertise, particularly through multimodal approaches, is expected to play a pivotal role in achieving truly personalized cancer care. As these technologies mature and gain broader validation by multicenter trials, they will become the cornerstone of precision oncology [50,51].

The omission of SLNB in the future could be due to advances in high-resolution imaging that could become sensitive enough to reliably detect micrometastases and most probably liquid biopsies for detecting molecular markers, which holds significant promise for the management of metastatic melanoma by enabling real-time, minimally invasive monitoring of treatment response, relapse, and disease progression [52]. The broad adoption of liquid biopsies testing in routine clinical practice depends on the current lack of standardization for sample collection, processing and analysis, and the necessity of clearly demonstrating the clinical utility because many assays have not yet been thoroughly validated in diverse patient populations. Beyond these methodological issues, a biological constraint exists in patients whose tumors release minimal or undetectable amounts of circulating tumor DNA (“non-shedders”), making it difficult to obtain reliable diagnostic, prognostic, or treatment-monitoring information from their samples [52,53].

## 4. Conclusions

Sentinel lymph node biopsy has unquestionably modified the clinical landscape of melanoma care, providing a minimally invasive yet highly precise method for nodal staging. Its journey, from a theoretical framework to an indispensable clinical tool, underscores the importance of innovation, evidence-based validation, and multidisciplinary collaboration in advancing surgical oncology.

Yet, despite its current indispensability, SLNB is at a crossroads. On one hand, technological advancements such as ICG fluorescence imaging, augmented reality, and AI-powered navigation systems promise to further enhance its accuracy and safety. On the other, the rapid evolution of systemic therapies and molecular diagnostics may ultimately reduce the reliance on surgical nodal evaluation. High-resolution imaging and tumor-specific biomarkers offer the possibility of real-time disease monitoring and individualized therapeutic strategies that could eventually replace surgical staging.

The future of SLNB lies not only in its refinement but also in its potential reinvention. As we stand at the threshold of a new era in oncology, the trajectory of SLNB exemplifies the broader shift toward data-driven, minimally invasive and patient-centered cancer care. Ultimately, the future of SLNB in melanoma lies in further refining patient selection criteria, integrating advanced prediction tools, expanding understanding of its therapeutic potential and exploring innovative treatment options that balance optimal oncologic outcomes with minimal patient morbidity.

## Figures and Tables

**Figure 1 medicina-61-01542-f001:**
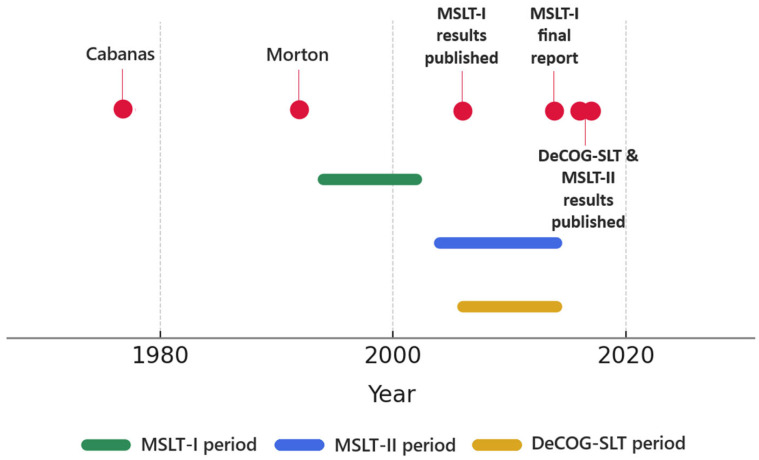
SLNB Milestones.

**Table 1 medicina-61-01542-t001:** Comparison of SLNB in cutaneous melanoma milestone trials.

Trial	Study Population	Main Survival Outcome	Disease-Free Survival (DFS)/Distant Metastasis-Free Survival (DMFS)
MSLT-I	intermediate (1.20–3.50 mm) and thick (>3.50 mm) cutaneous melanoma	No significant difference in 10-year melanoma-specific survival in overall population; benefit in SLN-positive intermediate-thickness group (HR 0.56; *p* = 0.006)	Intermediate thickness: DFS 71.3% vs. 64.7% (HR 0.76; *p* = 0.01); Thick: DFS 50.7% vs. 40.5% (HR 0.70; *p* = 0.03)
MSLT-II	SLN-positive; median follow-up 43 months	No improvement in 3-year melanoma-specific survival (86% vs. 86%; *p* = 0.42)	DFS slightly higher with CLND at 3 years: 68% vs. 63% (*p* = 0.05), due to improved regional control (92% vs. 77%; *p* < 0.001)
DeCOG-SLT	SLN-positive; median follow-up 72 months	No difference in 5-year overall survival (HR 0.99; *p* not significant)	5-year DMFS: 67.6% vs. 64.9% (HR 1.08; *p* = 0.87); 5-year RFS: HR 1.01

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
