# Peer review of "The Journey of Sentinel Lymph Node Biopsy in Cutaneous Melanoma: A Brief Narrative Review from Scalpel to Smart Tech"

_medicina, 2025, doi:10.3390/medicina61091542_

Round 1
Reviewer 1 Report
Comments and Suggestions for Authors
This review provides a comprehensive overview of the evolution, current standards, and future directions of sentinel lymph node biopsy (SLNB) in cutaneous melanoma. The authors effectively trace the historical development of SLNB, its prognostic significance, and the impact of technological advancements (e.g., indocyanine green fluorescence, augmented reality) and systemic therapies on its role. The manuscript is well-structured, clinically relevant, and identifies key gaps in knowledge, such as the need for standardized protocols and long-term efficacy data.
The review excellently contextualizes SLNB’s development, from Herbert Snow’s early observations to modern refinements (e.g., MSLT-I/II trials). Clear synthesis of SLNB indications (e.g., Breslow thickness >0.8 mm) and controversies (e.g., omission of CLND). Thoughtful discussion of emerging technologies (ICG, AR, AI) and systemic therapies, highlighting SLNB’s evolving role.
Nevertheless, the text-heavy format would benefit from tables/figures (e.g., timeline of SLNB milestones, comparison of tracer techniques). While future technologies (e.g., liquid biopsies) are mentioned, their current limitations (e.g., sensitivity for micrometastases) could be explored deeper. The variability in SLNB protocols (e.g., dual-tracer vs. ICG) warrants a dedicated section on unresolved standardization challenges.
The review identifies SLNB’s crossroads between refinement and potential obsolescence due to systemic therapies but could explicitly address how interim solutions (e.g., neoadjuvant trials like PRADO) might bridge this gap. Recent reviews (e.g., Cheng et al., 2023) are cited, but this manuscript distinguishes itself by integrating AR/AI advancements.
In addition I would like to put some specific comments:
Lines 90–100: Clarify whether Morton’s 1992 study was the first to apply SLNB to melanoma or built on prior work (e.g., Cabanas’ penile cancer studies).
Line 122: MSLT-I’s "clear benefit" in recurrence-free survival could be quantified (e.g., hazard ratios) for clarity.
Lines 180–185: The 10% risk threshold for SLNB is useful but could reference validation studies (e.g., Wong et al., 2005 nomogram).
Line 200: ICG’s cost-effectiveness claim needs supporting data (e.g., Kwizera et al., 2023).
Lines 240–250: AR’s "promising results" in oral cancer (von Niederhäusern et al.) should note scalability challenges (e.g., registration errors).
Line 265: Liquid biopsies’ promise could contrast with current sensitivity limitations (e.g., Sheriff et al., 2025).
Thus, the manuscript is logically organized but would benefit from subheadings (e.g., "Technological Innovations" under Future Directions). Hypotheses (e.g., AR’s utility) are plausible but require more critical appraisal of limitations. SLNB protocols (e.g., tracer doses) are described but could be summarized in a table for clinical utility. Consistent with evidence but could explicitly call for multicenter trials to validate AR/AI tools.
Author Response
Comment 1: Nevertheless, the text-heavy format would benefit from tables/figures (e.g., timeline of SLNB milestones, comparison of tracer techniques). While future technologies (e.g., liquid biopsies) are mentioned, their current limitations (e.g., sensitivity for micrometastases) could be explored deeper. The variability in SLNB protocols (e.g., dual-tracer vs. ICG) warrants a dedicated section on unresolved standardization challenges.
Response 1: We agree, therefore we added a timeline of important SLNB milestones. Liquid biopsies are an important and actual topic which we extended after review and with the upcoming improving in the medical field we hope to have less limitations. The differences in SLNB techniques was a good addition we added to the review at your advice, making a dedicated section for that.
Comment 2: Lines 90–100: Clarify whether Morton’s 1992 study was the first to apply SLNB to melanoma or built on prior work (e.g., Cabanas’ penile cancer studies).
Response 2: Morton's work was first to apply SLNB to melanoma. Dr. Morton didn't mentioned Cabanas in his paper, so we are not sure if its based in his work or not, but Cabanas is worth mentioning because he was the first to publish SLNB principles.
Comment 3: Line 122: MSLT-I’s "clear benefit" in recurrence-free survival could be quantified (e.g., hazard ratios) for clarity.
Response 3: We agree, therefore we made a table containing this data, and also comparing MSLT-II and DeCOG-SLT
Comment 4: Lines 180–185: The 10% risk threshold for SLNB is useful but could reference validation studies (e.g., Wong et al., 2005 nomogram).
Response 4: Wong et al., 2005 is cited in the last paragraph of that section [35]
Comment 5: Line 200: ICG’s cost-effectiveness claim needs supporting data (e.g., Kwizera et al., 2023).
Response 5: Kwizera et al., 2023 is cited in that paragraph [40]
Comment 6: Lines 240–250: AR’s "promising results" in oral cancer (von Niederhäusern et al.) should note scalability challenges (e.g., registration errors).
Response 6: We agree, AR technology is just emerging and a lot of challenges and errors could influence the operability of this technique. We also mentioned the limitations in the next paragraph.
Comment 7: Line 265: Liquid biopsies’ promise could contrast with current sensitivity limitations (e.g., Sheriff et al., 2025).
Response 7: We dealt with this subject at the first Comment
Reviewer 2 Report
Comments and Suggestions for Authors
Thorough narrative review that tracks the history, current standards, and future directions of SLNB in cutaneous melanoma. Suggestions:
- Condense repetitive historical descriptions and expand on emerging trends (e.g., AI in pathology, liquid biopsy validation, and cost-effectiveness analysis).
- MSLT-I and MSLT-II findings are stated but not dissected in terms of statistical power, follow-up, or clinical implications. Maybe do a comparative analysis?
- When comparing studies, clarifying study design, sample size and evidence level will be helpful.
-
Adding a timeline figure of SLNB milestones and maybe summarizing SLNB indications by guideline will be helpful.
- Some historical references are very old (Snow, 1892). Consider citing more recent papers.
Author Response
Comment 1: Condense repetitive historical descriptions and expand on emerging trends (e.g., AI in pathology, liquid biopsy validation, and cost-effectiveness analysis).
Response 1: We agree, and we expanded on emerging trends (e.g. liquid biopsy)
Comment 2: MSLT-I and MSLT-II findings are stated but not dissected in terms of statistical power, follow-up, or clinical implications. Maybe do a comparative analysis?
Response 2: We agree, and we made a comparison Table of MSLT-I, MSLT-II and DeCOG-SLT
Comment 3: When comparing studies, clarifying study design, sample size and evidence level will be helpful.
Response 3: We added a table of comparison for the major trials
Comment 4: Adding a timeline figure of SLNB milestones and maybe summarizing SLNB indications by guideline will be helpful.
Response 4: We agree. It was a great idea that improved the design and the content of the article
Comment 5: Some historical references are very old (Snow, 1892). Consider citing more recent papers.
Response 5: The majority of the papers are relatively new, but we chose to cite even very old articles like Snow's and Cabanas because it is a review that traces the history of SLNB, so they should be part of it